# Raloxifene Protects Oxygen-Glucose-Deprived Astrocyte Cells Used to Mimic Hypoxic-Ischemic Brain Injury

**DOI:** 10.3390/ijms252212121

**Published:** 2024-11-12

**Authors:** Nicolás Toro-Urrego, Juan P. Luaces, Tamara Kobiec, Lucas Udovin, Sofía Bordet, Matilde Otero-Losada, Francisco Capani

**Affiliations:** 1Centro de Altos Estudios en Ciencias Humanas y de la Salud, Universidad Abierta Interamericana, Consejo Nacional de Investigaciones Científicas y Técnicas, CAECIHS, UAI-CONICET, Buenos Aires C1270AAH, Argentina; nicolas.toro3@gmail.com (N.T.-U.); juanpluaces@yahoo.com (J.P.L.); tamara.kobiec@gmail.com (T.K.); lucas2304@hotmail.com (L.U.); sofia.bordet@gmail.com (S.B.); 2Centro de Investigaciones en Psicología y Psicopedagogía (CIPP), Facultad de Psicología y Psicopedagogía, Pontificia Universidad Católica Argentina (UCA), Buenos Aires C1107AFB, Argentina; 3Instituto de Ciencias Biomédicas, Facultad de Ciencias de la Salud, Universidad Autónoma de Chile, Santiago 7500912, Chile

**Keywords:** raloxifene, neuroprotection, astrocytes, glucose deprivation, oxygen deprivation, hypoxic–ischemic brain injury

## Abstract

Perinatal asphyxia (PA) is a clinical condition characterized by oxygen supply suspension before, during, or immediately after birth, and it is an important risk factor for neurodevelopmental damage. Its estimated 1/1000 live births incidence in developed countries rises to 5–10-fold in developing countries. Schizophrenia, cerebral palsy, mental retardation, epilepsy, blindness, and others are among the highly disabling chronic pathologies associated with PA. However, so far, there is no effective therapy to neutralize or reduce PA-induced harm. Selective regulators of estrogen activity in tissues and selective estrogen receptor modulators like raloxifene have shown neuroprotective activity in different pathological scenarios. Their effect on PA is yet unknown. The purpose of this paper is to examine whether raloxifene showed neuroprotection in an oxygen–glucose deprivation/reoxygenation astrocyte cell model. To study this issue, T98G cells in culture were treated with a glucose-free DMEM medium and incubated at 37 °C in a hypoxia chamber with 1% O_2_ for 3, 6, 12, and 24 h. Cultures were supplemented with raloxifene 10, and 100 nM during both glucose and oxygen deprivation and reoxygenation periods. Raloxifene 100 nM and 10 nM improved cell survival—65.34% and 70.56%, respectively, compared with the control cell groups. Mitochondrial membrane potential was preserved by 58.9% 10 nM raloxifene and 81.57% 100 nM raloxifene cotreatment. Raloxifene co-treatment reduced superoxide production by 72.72% and peroxide production by 57%. Mitochondrial mass was preserved by 47.4%, 75.5%, and 89% in T98G cells exposed to 6-h oxygen–glucose deprivation followed by 3, 6, and 9 h of reoxygenation, respectively. Therefore, raloxifene improved cell survival and mitochondrial membrane potential and reduced lipid peroxidation and reactive oxygen species (ROS) production, suggesting a direct effect on mitochondria. In this study, raloxifene protected oxygen–glucose-deprived astrocyte cells, used to mimic hypoxic–ischemic brain injury. Two examiners performed the qualitative assessment in a double-blind fashion.

## 1. Introduction

Perinatal asphyxia (PA) is a clinical syndrome characterized by an oxygen supply shortage around birth time [1]. It may be the result of a variety of factors, including umbilical cord compression, alterations in gas exchange in the placenta, and fetal pulmonary failure, resulting in multiorgan oxygen (hypoxia) and/or hypoperfusion (ischemia) during the perinatal period [2].

PA affects not only glial and neural circuits in the corpus striatum and the hippocampus [3,4] but also increases presynaptic terminal button density and striatal and cortical dendritic spines [5,6]. It also leads to necrosis and apoptosis in these areas and the cortex [7] and astrogliosis in the striatum and cerebellum [4,8]. It disrupts the dopaminergic, GABAergic, glutamatergic, and redox systems [7,9,10] leading to oxidative stress [9] and protein aggregation with ubiquitination [5], among other deleterious effects.

Mortality caused by PA has decreased in the last decades, while morbidity has increased [11]. Many physical and mental disorders can be associated with PA [12], including cognitive deficits [8,13,14], seizures and/or epilepsy [15], schizophrenia [16], autism spectrum disorders [17], attention-deficit/hyperactivity [18], and neurodegenerative disorders [19].

Glucose is the primary energy source for the adult brain, with neurons having the highest energy demand among cells [20]. From glucose metabolism, the brain gets precursors for neurotransmitter synthesis, ATP for physiological functions, and neuronal and non-neuronal cell maintenance [21]. However, astrocytes and other brain cells are crucial for brain glucose metabolism [22].

Astrocytes make up nearly 25% of brain volume, and their role in multiple processes has been studied over the past 20 years, including supporting the nervous system, maintaining the cerebral microenvironment for proper function, and regulating cerebral blood flow, which is crucial for optimal neuronal function. One of their most important functions is their involvement in brain metabolism. On the one hand, astrocytes take up glucose from the blood vessels and provide energy metabolites to neurons [23]. Through the astrocyte-neuron lactate shuttle (ANLS), astrocytes supply neurons with lactate, which serves as a substrate for the citric acid cycle to meet their energy demands [24].

Neurons are very sensitive to energy deficits. Changes in glucose metabolism have been linked to cell death pathways and autophagy, increasing the risk of various central nervous system disorders. Different studies have shown how these alterations are associated with various neurodegenerative disorders’ progression, including Alzheimer’s disease, Parkinson’s disease, amyotrophic lateral sclerosis, neuroglycopenia, and Huntington’s disease, among others [21,25]. Reduced oxidation levels observed in post-ischemic brain tissues suggest these alterations may contribute to cerebrovascular and neurodegenerative abnormalities, including stroke [25].

Therefore, there is considerable research focused on the prevention or reversal of neuronal damage caused by hypoglycemia. Steroids, such as estrogens and progesterone, have shown the ability to modulate neuronal activity and provide neuroprotective effects against toxicity and neurodegeneration in vivo studies [26]. Neurons and glial cells have high-affinity steroid receptors, highlighting their importance in brain activity [26,27,28]. It has been described that the protective response of astrocytes can be modulated by neurosteroids, which regulate and modulate the expression of many genes involved in the development, connectivity, and survival of the nervous system in both glial and neuronal cells [29,30].

Raloxifene is a benzothiophene-derived selective estrogen receptor modulator (SERM) that was developed by Eli Lilly under the brand name “Evista” and approved by the U.S. Food and Drug Administration (FDA) to treat osteoporosis in menopause more than twenty years ago [31]. Later on, studies were conducted on its possible application to neurological diseases and brain injuries [32].

Raloxifene modulates the morphology and functions of astrocytes, neurons, and microglia via its interaction with estrogen receptors α (ERα) and β (ERβ) and G-protein-coupled estrogen receptor (GPR30) activation [33,34].

Selective estrogen receptor modulators (SERMs) exert brain protection by reducing inflammatory responses, lipid peroxidation, the production of reactive oxygen species (ROS), astrogliosis, and leukocyte infiltration in the injured area. These beneficial effects improve spatial learning and memory and can promote neuronal survival [33]. Raloxifene treatment increases neuronal survival because of its anti-inflammatory effect. Improved learning and memory and less cognitive impairment were reported in elderly women and cancer patients after raloxifene treatment [33,35].

Raloxifene preserved neurogenesis in the cerebral cortex and spinal cord in a middle cerebral artery occlusion model [36]. In male rodents treated with tamoxifen and raloxifene, hippocampal pyramidal neurons showed increased dendritic spine density and geometry changes—likely mediated by increased BDNF—[37]. Raloxifene might be involved in brain plasticity, learning processes, and long-term memory, and yet more studies are needed in this regard [33].

Estrogen affected ERα expression and protected astrocytes under oxygen–glucose deprivation and hypoxic injury [38], likely involving hypoxia-inducible factor-1 (HIF-1). This factor has a vital role in cellular adaptation to hypoxic conditions, is expressed in astrocytes under ischemia, and is associated with glutathione (GSH) increase, astrocytic survival, and protection against glutamate toxicity [39]. In a model of oxygen–glucose deprivation in cortical neurons, raloxifene also prevented apoptosis and necrosis by binding to GPR30, which triggers rapid intracellular calcium signaling and antiapoptotic neuroprotective mechanism activation [40]. Under oxygen–glucose deprivation, raloxifene induced the activation of transcription factor Nrf2 and antioxidant response element (ARE), an ER-independent promoter of redox homeostasis in neurons [40]. ARE activation is involved in the expression of protective genes against oxidative stress and depends on Nrf2 activity [41]. Evidence shows that SERMs like raloxifene have antioxidant potential, antiapoptotic properties, and trigger survival and differentiation mechanisms in neurons and glia. Thus, SERMs might mitigate brain damage [33].

## 2. Results

### 2.1. Cell Viability

The MTT assay was used to examine the effect of raloxifene on OGD/R-exposed T98G cells’ viability. Figure 1A shows cell viability when cells were co-treated with OGD and raloxifene at different concentrations (during injury). Cells were treated with raloxifene since they were subjected to oxygen and glucose deprivation until the end of the reoxygenation period. Cells exposed to OGD for 6 h followed by reoxygenation for 3 h showed 65.34% (*p* = 0.0021) and 70.56% (*p* < 0.0001) recovery with 100 nM and 10 nM of raloxifene, respectively, compared with the untreated control cells (Figure 1A).

In this model, cell viability assessed with MTT decreased from the early hours of OGD/reoxygenation, progressively decreasing from 3 h to 24 h of OGD/24 h reoxygenation. Approximately 50% of the damaged cells died after 24 h OGD/24 h reoxygenation.

Cells’ viability and morphology are directly related. We performed a qualitative analysis using a phase contrast microscope to evaluate cell shape changes after the OGD/R insult in T98G cells (Figure 1B–F). The OGD/R insult resulted in cellular shrinkage, vacuolization, and a decrease in the number of processes (Figure 1C). Co-treatment with raloxifene decreased cell degeneration (Figure 1D,F), showing astrocytic-like morphology comparable to control cells (Figure 1B).

Our next aim was to analyze the average cell area of randomly chosen cells under our experimental conditions. We noted a qualitative change in cell morphology during glucose deprivation (OGD/R) stress (Figure 1C). Post-raloxifene co-treatment, morphological changes were reduced, returning to their control levels. To quantify morphological changes, we measured the mean cell area across all experimental conditions. Raloxifene-treated cells’ area (450 ± 35 µm) was comparable to controls’ (460 ± 22 µm) (*p* < 0.001). Cell area decreased in the OGD/R group compared with the control group (311 ± 37 µm) (*p* < 0.001).

### 2.2. Raloxifene-Reduced Superoxide Production in OGD-Exposed T98G Cells

Flow cytometry analysis using dihydroethidium (DHE) was used to measure the effect of treatment with raloxifene on superoxide anion (O 2^•−^) production in OGD-exposed cells. Dihydroethidium (DHE) is a cell-permeable compound that interacts with the superoxide anion O 2^•−^ to form ethidium, which binds to nucleic acids and emits bright red fluorescence [42]. (Figure 2, Appendix A).

We measured hydrogen peroxide production to confirm a decrease in reactive oxygen species (ROS) production by raloxifene in OGD/R-exposed T98G cells using another dye. DCFDA (Dichlorofluorescein diacetate) is an organic dye of the fluorescein family that emits fluorescence after oxidation by hydrogen peroxide and other ROS [43]. Using DCFDA, the mean fluorescence intensity was measured in cell culture microphotographs (Figure 3).

Figure 3 illustrates fluorescence intensity levels after 6 h of OGD insult and 6 h of reoxygenation (Figure 3E). H_2_O_2_ production increased in OGD/R-exposed cells compared with control cells (*p* < 0.0001) (Figure 3A,B). Cotreatment with raloxifene attenuated H_2_O_2_ production (Figure 3C,D).

### 2.3. Raloxifene Effect on Mitochondrial Membrane Potential Loss in Reperfused OGD-Exposed T98G Cells

To evaluate the effect of raloxifene on mitochondrial membrane potential (Δψm), a quantitative analysis was performed using flow cytometry (Figure 4A, Appendix A). Cells treated with 10 nM of the uncoupling protonophore carbonyl cyanide m-chlorophenyl hydrazone (CCCP, Sigma, St Louis, MI, USA) were used as a mitochondrial oxidative phosphorylation inhibition experimental control. CCCP dissipates the mitochondrial membrane potential, providing a baseline for analysis. Δψm was measured using the lipophilic cation tetramethylrhodamine methyl ester (TMRM), a dye that changes its intensity in response to Δψm changes [44]. 4 shows OGD/R and raloxifene effects on T98G cells’ Δψm. Mitochondrial membrane potential loss (*p* = 0.0006) was observed after 6 h of OGD followed by 3 h of reoxygenation and was attenuated by 10 nM raloxifene treatment (*p* = 0.0180) compared with untreated OGD/R-exposed cells (Figure 4A). Fluorescence microphotographs showed a similar pattern with decreased fluorescence staining after OGD/R exposure and intensity recovery after raloxifene treatment (Figure 4B–E, Appendix A).

### 2.4. Raloxifene-Attenuated Mitochondrial Mass Reduction in OGD/R-Exposed T98G Cells

To determine the OGD/R effect on mitochondrial mass, a quantitative analysis was performed using nonyl acridine orange (NAO) and flow cytometry (Figure 5). NAO is a marker of the mitochondrial membrane surface and measures mitochondrial lipid peroxidation by detecting cardiolipin oxidation [45].

OGD/R resulted in a mitochondrial mass reduction in OGD/R-exposed T98G cells compared with control cells (*p* < 0.0001) exposed to 6 h OGD and 3 h reoxygenation (Figure 5O). This change was counteracted by co-treatment with 100 nM (*p* = 0.0349) and 10 nM (*p* = 0.0038) raloxifene, which preserved mitochondrial mass (Figure 5O). Fluorescence microphotographs confirmed a similar pattern using the NAO dye. OGD/R exposure decreased fluorescence intensity compared with control cells, while treatment with raloxifene preserved fluorescence intensity (Figure 5A–D).

To understand lipid peroxidation dynamics, mitochondrial mass changes were measured after 6 h of OGD using fluorescence microphotographs, followed by different reoxygenation periods (Figure 5D,H,L). Figure 5 shows intensity loss in OGD/R-exposed cells and the raloxifene corrective effect at different reoxygenation times between 6 h (Figure 5H) and 9 h (Figure 5L).

Qualitative studies measuring fluorescence intensity in the microphotographs confirmed what was observed (Figure 5). An amount of 10 nM (*p* = 0.0073) and 100 nM (*p* = 0.0468) raloxifene attenuated mitochondrial mass loss at different reoxygenation times (6 and 9 h, respectively) in 6 h OGD-exposed cells (Figure 5H,L).

## 3. Discussion

This study focused on characterizing the astrocytic T98G-OGD/R model. Careful analysis of the events occurring during OGD offered noteworthy evidence regarding cell viability, lipid peroxidation, reactive species production, and mitochondrial function.

In vitro experiments are crucial given the methodological limitations of in vivo models [46]. The choice of an in vitro model that meets the specific requirements of the parameters to be evaluated is of paramount importance. Models using hypoxia-simulating agents are among the in vitro approaches. These are based on producing, at the molecular level, the effects caused by low oxygen concentrations, primarily those based on hypoxia-inducible factor 1α (HIF1A) expression [46]. However, they do not consider removing glucose, the key energy supply for brain metabolism.

Estrogens have been reported to be neuroprotective in astrocytes [47,48,49,50]. Raloxifene neuroprotection has been observed in different brain injury models: schizophrenia, excitotoxic neuronal death [50], and glucose deprivation [51]. Among others, raloxifene is a neurosteroid that can modify various metabolic and genomic responses in the brain and has shown neuroprotective effects in different pathologies [50,51,52]. Neuroprotective effects have been observed in murine models of common carotid artery ligation, where neurogenesis in the ipsilateral subventricular zone has been observed in rats treated with raloxifene [36]. Raloxifene treatment has shown neuroprotective effects, mitigating cell death. An amount of 100 nM of raloxifene has protected against metabolic insults and cerebral injury in astrocytic models with T98G cells [51]. The present findings agree with these observations. Cotreatment with 100 nM raloxifene reduced cell death after OGD-reoxygenation (Figure 1). Different molecular mechanisms may mediate cell death decreases [21,26,53,54,55,56], which may explain the results observed in the study.

In astrocytic models, raloxifene regulates glutamate transporters (GLAST and GLT-1), improving cell survival [50]. Noteworthy, its protective effects might involve expressing the anti-apoptotic gene Bcl-2 [57]. Through non-genomic mechanisms, raloxifene can activate survival proteins like MAPk, PI3K/Akt, Src, and CREB, related to anti-apoptotic metabolic pathways and neuroprotection [50,57,58].

Changes in cell morphology have been associated with cell viability, in particular, the state in which the cell is found. Cells exhibit different characteristics during apoptosis, including nuclear envelope disintegration, cytoplasmic condensation, and surface reduction [59,60], parameters mostly affecting cell morphology. These changes align with the findings in this study (Figure 1B–D), where raloxifene decreases morphological alterations caused by glucose and oxygen deprivation, showing a potential reduction in apoptotic cellular processes.

In our model, superoxide and hydrogen peroxide production was detected from 6 h of OGD/R injury onwards (Figure 2 and Figure 3). In physiological conditions, these species are produced in regulated concentrations and are transformed into less reactive species through energy-dependent mechanisms [61,62,63,64]. An energy shortage hinders the transformation of these species into a less reactive form, triggering damage to cellular machinery. Therefore, this parameter is a potential indicator of damage that, if reversed, could explain neuroprotective effects. ROS production depends on Δψm [65,66]. Maintaining an optimal mitochondrial membrane potential (maximum 140 mV) prevents ROS formation and exhaustively uses ATP production capacity.

Reactive oxygen species can be generated at low Δψm levels under certain conditions. Yet, ROS are produced in excess when mitochondrial membrane potential levels are high and Δψm exceeds 140 mV (membrane hyperpolarization). In these conditions, ROS production in mitochondrial respiratory complexes I (NADH: ubiquinone oxidoreductase) and III (ubiquinol–cytochrome c oxidoreductase) increases exponentially [67], and an increase in oxidative stress could generate significant morphological changes. This corresponds to the results presented in this study, where an exponential increase in superoxide anion production at hour 6 of OGD/R (Figure 2) and H_2_O_2_ at hour 6 (Figure 3D) is observed. These periods match the highest peaks of mitochondrial membrane potential production in glucose deprivation insults in T98G cells, as reported in previous studies [29]. This phenomenon is attributed to the mitochondria’s need to adapt to the cell’s energy needs, regardless of the deleterious increase in mitochondrial membrane potential. The cell enters mitochondrial apoptosis, characterized by excessive calcium release, Δψm hyperpolarization, and exacerbated ROS production [68,69].

The collected data revealed a significant alteration in mitochondrial potential at 6 h of OGD and 3 h of reoxygenation (Figure 4). This loss of potential coincides with the loss of viability described earlier. In different models, this potential has been found to increase to a peak within the first few hours of damage and then decrease gradually until a complete loss, matching cell death time [29,51].

While hydrogen peroxide is less reactive, it can form different reactive hydroxyl radicals in the presence of iron ions, starting lipid peroxidation cascades in the cell membrane. This oxidative stress is a major problem in neurodegenerative diseases and one of their major causes [69,70]. Neurosteroids have shown the ability to attenuate these oxidative stress processes, decreasing ROS levels in injuries caused by oxidative stress in particular [71,72]. This aligns with the results reported by Capani et al. [73]. In a perinatal asphyxia model, they found increased H_2_O_2_ production—likely related to an increase in the dynamics of superoxide transfer and transformation—preventing the harmful accumulation of this more dangerous species. Superoxide species conversion to less reactive forms like hydrogen peroxide and water helps cells achieve a more stable physiological state during injury [72,74]. However, this ROS increase, like membrane hyperpolarization, is not sustained throughout the injury period. As cellular energy levels have decreased because of a cellular energy shortage, the cell has shifted to other energy sources like fatty acid oxidation or glycogenolysis in the early hours of insult [75], which may explain the changes in ROS production in this study.

The mitochondrial role in cell death depends on the control of energy metabolism, ROS production, and the release of apoptotic factors into the cytoplasm. Cytochrome C is the most prominent pro-apoptotic factor [19], and mitochondrial apoptosis-inducing proteins like endonuclease G, Smac/DIABLO, and Omi/HtrA have also been described to play a significant role in apoptosis regulation. These pro-apoptotic factors are not necessarily released through mitochondrial permeability transition pores (MPT). This suggests that changes in Δψm are directly related to cellular necrosis and apoptosis [76]. Our results showed raloxifene preserved mitochondrial membrane potential in cells exposed to oxygen and glucose deprivation, in agreement with Vesga-Jiménez et al. [51], who reported that raloxifene protected astrocytes from oxidative stress.

One of the most critical targets is cardiolipin, a phospholipid found in the inner mitochondrial membrane. It is crucial for the insertion into the membrane and the function of cytochrome C, cytochrome C oxidase, and other phosphorylation complexes. It is required for the optimal functioning of complex I (NADH: ubiquinone reductase), complex III (NADH: ubiquinone cytochrome C oxidoreductase), complex IV (cytochrome C oxidase), and complex V (ATP synthase). Changes in phospholipid structure can lead to mitochondrial dysfunction, as the integrity of cardiolipin depends on it. However, its high content of fatty acids makes it susceptible to ROS-induced damage [77,78,79]. Considering the above, the findings in this study may help understand the reasons behind the mitochondrial dysfunction observed.

Figure 5 shows how OGD/R affected T98G cells’ mitochondrial mass at different insult-exposure times. Raloxifene preserved cells from mitochondrial mass loss, disclosing another edge of its neuroprotective effects. The decrease in ROS production has been found to mediate this effect [79].

In sum, understanding whether and how OGD/R damage contributes to disease could explain why hypoxic–ischemic brain injury is a complex concurrence of simultaneous processes and how it may be approached to enhance the therapeutic effects of neuroprotective treatments. The outcome of this part of the study shows that raloxifene exerted neuroprotective effects on T98G astrocytic cells exposed to hypoxia-reoxygenation injury. The survival of these cells in the OGD/R model suggests astrocytes might participate in the protective effects of these neurosteroids in hypoxia-ischemia injury. Further studies will determine if the present findings can be extrapolated to an in vivo model.

## 4. Materials and Methods

### 4.1. T98G Cell Cultures

T98G cell line was used as an astrocytic cell model system (ATCC CRL-1690) [80,81,82]. Cells were kept under exponential growth in Dulbecco’s modified Eagle’s medium (DMEM) (LONZA, Walkersville, MD, USA), containing 10% fetal bovine serum (FBS, LONZA, Walkersville, MD, USA), and 10 U penicillin/10 μg streptomycin/25 ng amphotericin (LONZA, Walkersville, MD, USA). The medium was changed three times a week. Cultures were incubated at 37 °C in a humidified atmosphere containing 5% carbon dioxide and 95% oxygen. Cells were seeded in 96-well plates for cell death measurement, 12-well plates for flow cytometry determinations, and 24-well plates for tetra-methyl rhodamine methyl ester (TMRM) and fluorescence measurements and microphotographs. The mean cell area was assessed by analyzing black-and-white phase contrast microphotographs using ImageJ software, version 1.54j. Using ImageJ, the area of each randomly selected cell was determined, and the software was calibrated by measuring a known distance. The average cell area was determined for every experimental group, each measured in triplicate, with at least 25 cells analyzed per condition.

### 4.2. Drug Treatments

To determine raloxifene concentrations to be tested on T98G cells, dose-response curves were performed. Cells were trypsinized for 3 min at 37 °C, and DMEM medium was added for trypsin inactivation. The cells were transferred to a 15 mL centrifuge tube and centrifuged at 1700 rpm for 5 min. The supernatant was discarded, and the cell pellet was resuspended in 1 mL of medium. Around 100,000 cells/well were seeded in a 24-well plate with a final volume of 500 μL/well. The plate was incubated at 37 °C and 5% CO_2_ until reaching 80% confluence. Next, the cell medium was replaced with serum-free DMEM and incubated for 12 h. Once this time was completed, the cells were treated with a DMEM medium without glucose supplemented with raloxifene at concentrations of 10, and 100 nM during the metabolic insult hours of glucose and oxygen deprivation, as well as during the reoxygenation period. Neurosteroid solutions were prepared by diluting a 100 μM stock drug solution in DMSO in a glucose-free DMEM medium to achieve a concentration of 0.0001% of the vehicle.

### 4.3. Oxygen and Glucose Deprivation

To induce oxygen–glucose deprivation, T98G cells were first washed three times with a glucose-free DMEM solution, then treated with glucose-free DMEM and incubated at 37 °C in a hypoxia chamber (Stemcell) with 1% O_2_ for 3, 6, 12, and 24 h. Control cultures were treated with DMEM medium under normoxic conditions for the same incubation times. Then, the hypoxic cells were subjected to reoxygenation under normoxic conditions with both oxygen and glucose.

### 4.4. Cell Viability Assessment

Cell viability was assessed using the MTT (3-(4,5-dimethylthiazol-2-yl)-2,5-diphenyltetrazolium) assay (Sigma, St. Louis, MI, USA). Cells were seeded in 96-well plates in DMEM culture medium containing 10% fetal bovine serum at a seeding density of 10,000 cells per well and incubated for 2–3 days until reaching confluence. Subsequently, cells were treated according to different experimental schemes. Viability was assessed after oxygen and glucose deprivation/reoxygenation (OGD/R) by adding an MTT solution at a final concentration of 5 mg/mL for 4 h at 37 °C. Cells were then lysed by the addition of dimethyl sulfoxide (DMSO). The resulting blue formazan product was measured using a plate reader at 595 nm (spectrophotometer GloMax® Discover Multimode Microplate Reader, Promega Corporation, Madison, WI, USA). The values were normalized to the control cultures without oxygen and glucose deprivation, which were considered to have 100% cell survival. Each assay was performed with a minimum of 18 replicated wells for each condition, with three replicates per experiment. Cell morphology was qualitatively analyzed using a phase contrast microscope to evaluate cell shape changes after OGD/R insult in T98G cells (Fluorescence NIKON Eclipse Ti-E PF microscope, Tokyo, Japan).

### 4.5. Reactive Oxygen Species (ROS) Production Determination

Cells were seeded at a density of 75,000 cells per well into 12-well plates in DMEM culture medium containing 10% FBS and then treated according to the experimental procedure the next day. Hydrogen peroxide (H_2_O_2_) and superoxide production in cells exposed to OGD were measured using 2′,7′-Dichlorofluorescein Diacetate (DCFDA) at 1 nM and Dihydroethidium (DHE), respectively. Cells were incubated with the compounds at 37 °C for 30 min in the dark, washed with 1X PBS, and trypsinized for flow cytometry analysis (Becton Dickinson FACS Calibur cytometer, Franklin Lakes, NJ, USA). Each assay was performed with six replicates for each condition and 3 repetitions. The cells were observed using fluorescence microscopy, and photomicrographs were taken. The experiment was performed in quintuplicate.

### 4.6. Mitochondrial Membrane Potential Determination

Mitochondrial membrane potential was measured by flow cytometry using Tetramethyl Rhodamine Methyl Ester (TMRM). TMRM is a fluorescent cationic probe that permeates the cell and is taken up by active mitochondria [44]. Following treatment with neurosteroids and OGD/reoxygenation, T98G cells were incubated with a 500 nM TMRM solution in the absence of light at 37 °C for 20 min. After the incubation period, the probe was removed, and the cells were washed with 1X PBS three times to remove residual TMRM. Cells treated with 10 nM of the uncoupling protonophore carbonyl cyanide m-chlorophenyl hydrazone (CCCP, Sigma, St. Louis, MI, USA) were used as a mitochondrial oxidative phosphorylation inhibition experimental control. CCCP dissipates the mitochondrial membrane potential, providing a baseline for analysis. The quantitative analysis was evaluated by flow cytometry (Becton Dickinson FACS Calibur cytometer, Franklin Lakes, NJ, USA), and the cells were observed using fluorescence microscopy, and photomicrographs were taken. The experiment was performed in quintuplicate.

### 4.7. Mitochondrial Mass Determination

Mitochondrial mass was determined using Acridine Orange Nonyl (NAO) and quantitative analysis by flow cytometry and fluorescence microphotographs. After completion of the OGD insult and reoxygenation time, cells were washed three times with 1X PBS. Then, cells were trypsinized and transferred to centrifuge tubes with fetal bovine serum to inactivate the enzyme. The cells were centrifuged at 4500 rpm for 5 min and resuspended in a solution containing the light-protected NAO compound. Quantitative analysis was performed by flow cytometry (Becton Dickinson FACS Calibur cytometer) and fluorescence microscopy analysis. For 6 h OGD and 3 h reoxygenation conditions, flow cytometry was used, and for 6 h OGD and 3 h and 6 h reoxygenation conditions, fluorescence microscopy was used. For fluorescence microscopy analysis, cells were fixed with 4% paraformaldehyde (PFA) as described previously and stained with NAO. The cells were observed using fluorescence microscopy, and photomicrographs were taken. The experiment was performed in quintuplicate. The images were processed with Image J software, and the mean fluorescence intensity of randomly selected cells was determined as described below. The calculation of the mean fluorescence intensity of the cells was assessed using ImageJ. The microphotographs were opened in the software and pre-processed, eliminating the background. Then, 20 cells were randomly selected using a numbered grid in each microphotograph. The mean fluorescence value of the 20 cells was determined in 8 microphotographs for each treatment using the Measure algorithm of ImageJ and selecting each cell manually via ROI’s management. There were no variations in the conditions of the image processing. 

### 4.8. Statistical Analysis

The Kolmogorov–Smirnov and Levene’s tests were used to evaluate the normal distribution and homogeneity of variance, respectively. Data were examined by analysis of variance, followed by the post hoc Dunnet’s test for between-group comparisons and Tukey’s test for multiple comparisons. Data are expressed as the mean ± SEM. A statistically significant difference was set at *p* < 0.05. 

## Figures and Tables

**Figure 1 ijms-25-12121-f001:**
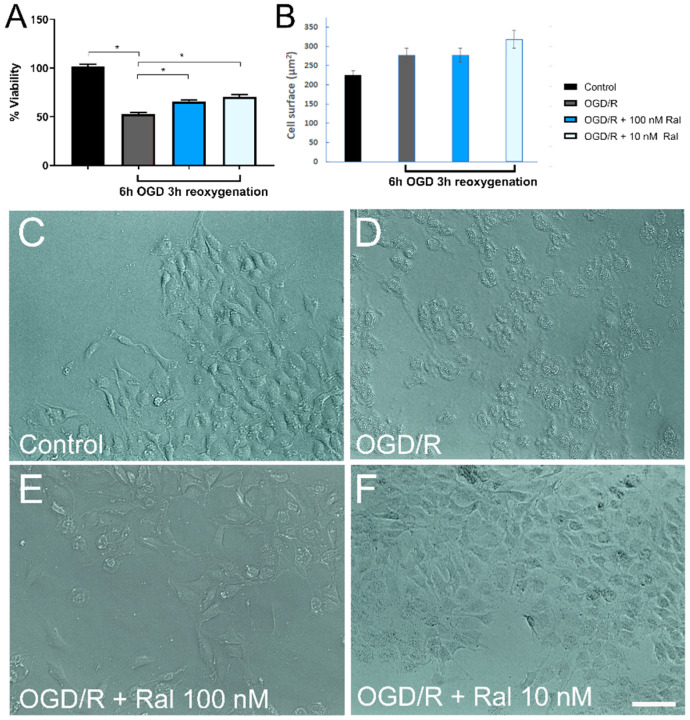
Raloxifene decreased OGD-induced cell death. (**A**) T98G cells were treated with different concentrations of raloxifene during 6 h of OGD and 3 h of reoxygenation, and cell viability was assessed by MTT assay. Data are represented as the mean ± SEM of four independent experiments. Control (101.99 ± 1.85); OGD/R (52.59 ± 2.02); OGD/R + 100 nM raloxifene (65.34 ± 2.03); OGD/R + 10 nM raloxifene (70.56 ± 2.36). Data were examined by analysis of variance, followed by the post hoc Dunnet’s test for between-group comparisons and Tukey’s test for multiple comparisons, * *p* < 0.005. (**B**) Cell surface quantification with different concentrations of raloxifene during 6 h of OGD and 3 h of reoxygenation. Data are represented as the mean ± SEM of four independent experiments. Control (225.3 ± 13.01); OGD/R (278.7 ± 18.51); OGD/R + 100 nM raloxifene (318.2 ± 21.86); OGD/R + 10 nM raloxifene (277.1 ± 18.16). Data were examined by analysis of variance, followed by the post hoc Dunnet’s test for between-group comparisons and Tukey’s test for multiple comparisons, * *p* < 0.005. (**C**–**F**) Raloxifene reduced morphological alterations induced by oxygen–glucose deprivation/reoxygenation. Representative microphotographs showing the morphology of cells exposed to (**C**) DMEM, (**D**) OGD/R, (**E**) OGD/R + Ral 100 nM, and (**F**) OGD/R + Ral 10 nM. Scale bar 50 µm.

**Figure 2 ijms-25-12121-f002:**
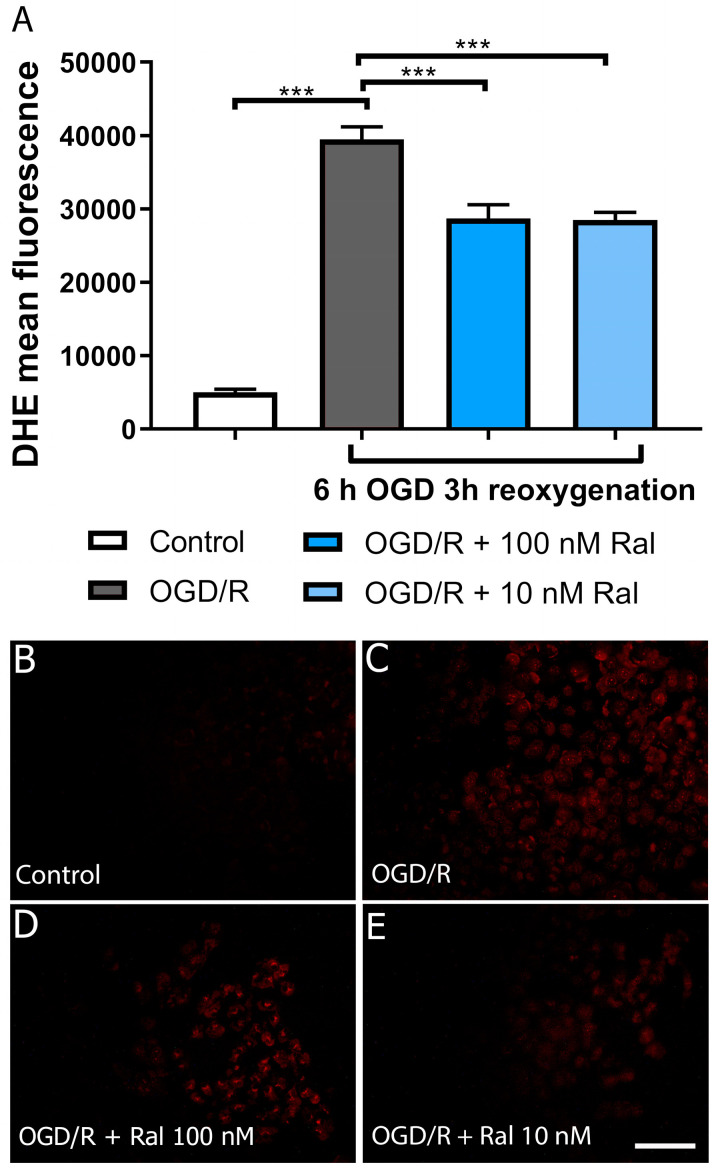
Raloxifene reduced superoxide production at 6 h of OGD and 3 h of reoxygenation. (**A**) Mean fluorescence values of dihydroethidium (DHE) intensity. (**B**–**E**) Representative fluorescence micrographs of dihydroethidium (DHE) staining in T98G cells exposed to (**B**) DMEM, (**C**) OGD/R, (**D**) OGD/R + Ral 100 nM with 6 h of OGD and 3 h of reoxygenation, and (**E**) OGD/R + Ral 10 nM with 6 h of OGD and 3 h of reoxygenation. *** *p* < 0.0001. Scale bar 50 µm.

**Figure 3 ijms-25-12121-f003:**
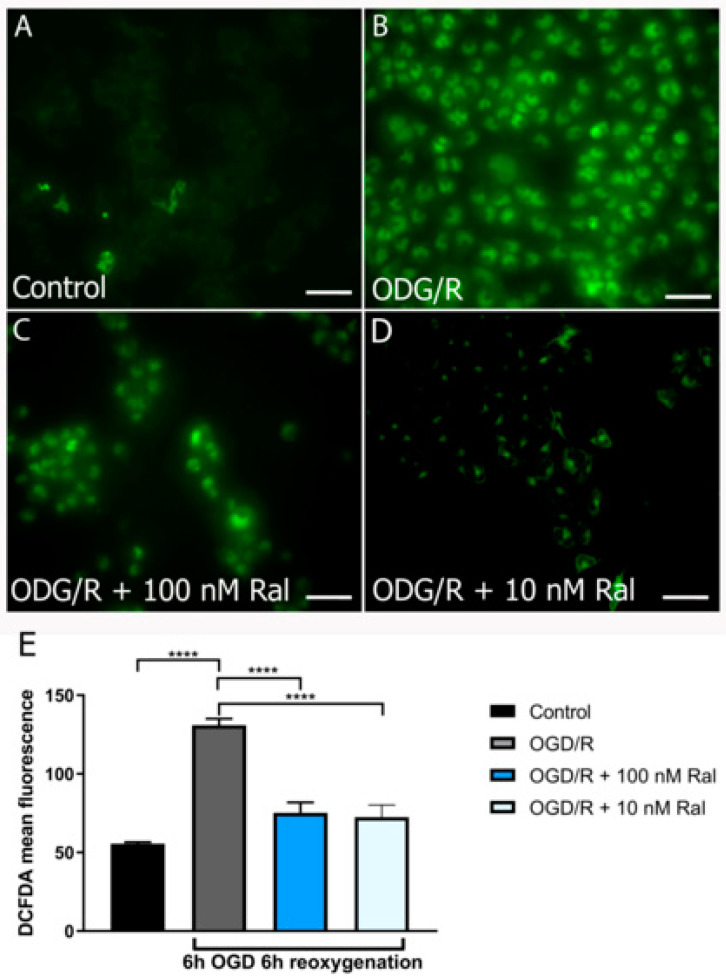
Raloxifene reduced peroxide production at 6 h of OGD and 6 h of reoxygenation. The figure shows the representative fluorescence microphotographs of 2′,7′-Dichlorofluorescin Diacetate (DCFDA) staining of T98G cells exposed to (**A**) Control, (**B**) OGD/R, (**C**) OGD/R, OGD/R + Ral 100 nM with 6 h of OGD and 6 h of reoxygenation, (**D**) OGD/R, OGD/R + Ral 10 nM with 6 h of OGD and 6 h of reoxygenation, and (**E**) the mean fluorescence values of DCFDA intensity measured by flow cytometry. Data are represented as the mean ± SEM of five independent experiments. Control (55.51 ± 1.03); OGD/R (131.00 ± 4.01); OGD/R + 100 nM raloxifene (75.15 ± 6.60); OGD/R + 10 nM raloxifene (72.38 ± 7.82). Data were examined by analysis of variance, followed by the post hoc Dunnet’s test for between-group comparisons and Tukey’s test for multiple comparisons **** *p* < 0.0001. Scale bar 50 µm.

**Figure 4 ijms-25-12121-f004:**
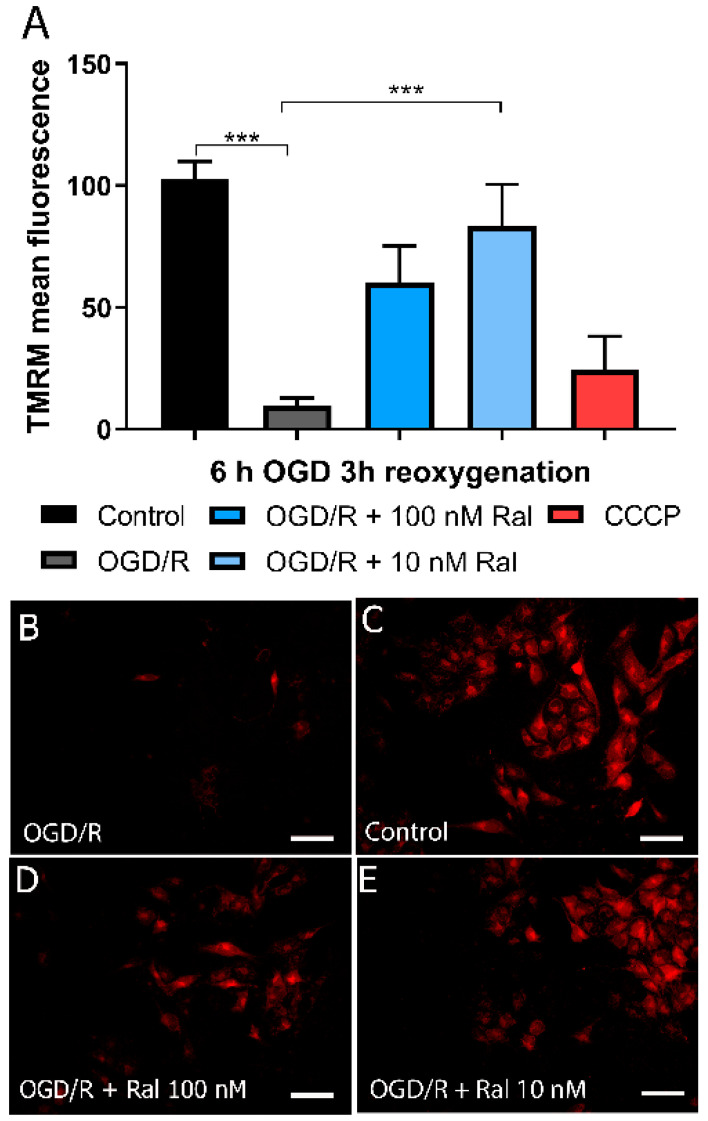
Raloxifene attenuated mitochondrial membrane potential loss at 6 h of OGD and 3 h of reoxygenation. (**A**) The figure shows the mean fluorescence values. (**B**–**E**) Representative fluores-cence micrographs of tetra-methyl rhodamine methyl ester (TMRM) staining in T98G cells exposed to (**B**) OGD/R, (**C**) DMEM, (**D**) OGD/R + Ral 100 nM with 6 h of OGD and 3 h of reoxygenation, and (**E**) OGD/R + Ral 10 nM with 6 h of OGD and 3 h of reoxygenation. *** *p* < 0.0001. Scale bar 50 µm.

**Figure 5 ijms-25-12121-f005:**
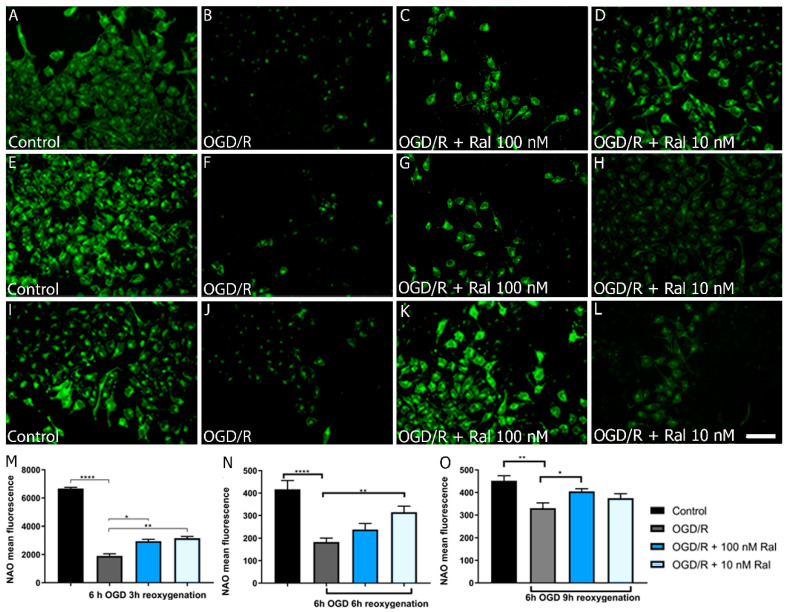
Raloxifene preserved mitochondrial mass in T98G cells exposed to 6 h of OGD and 3 h of reoxygenation. The figure shows the mitochondrial mass in T98G cells exposed to 6 h of oxygen–glucose deprivation (OGD) to 3 h (**A**–**D**), 6 h (**E**–**H**), and 9 h (**I**–**L**) of reoxygenation. The representative microphotographs of acridine orange (NAO) fluorescence in T98G astrocytic cells exposed to (**A**) DMEM, (**B**) OGD/R, (**C**) OGD/R + Ral 100 nM with 3 h of reoxygenation, and (**D**) OGD/R + Ral 10 nM with 3 h of reoxygenation. (**M**) Mean fluorescence values of NAO intensity in this period of insult. Data are represented as the mean ± SEM of five independent experiments. Control (6671.00 ± 86.18); OGD/R (1903.00 ± 155.30); OGD/R + 100 nM raloxifene (2940.00 ± 142.90); OGD/R + 10 nM raloxifene (3163.00 ± 119.80). (**E**) DMEM, (**F**) OGD/R, (**G**) OGD/R + Ral 100 nM with 6 h of reoxygenation, and (**H**) OGD/R + Ral 10 nM with 6 h of reoxygenation. (**N**) Mean fluorescence values of NAO intensity in this period of insult. Data are represented as the mean ± SEM of five independent experiments. Control (416.7.00 ± 39.47); OGD/R (183.1 ± 17.70); OGD + 100 nM raloxifene (238.4 ± 26.43); OGD + 10 nM raloxifene (314.6 ± 27.45) (**I**) DMEM, (**J**) OGD/R, (**K**) OGD/R + Ral 100 nM with 9 h of reoxygenation, and (**L**) OGD/R + Ral 10 nM with 9 h of reoxygenation. (**O**) Mean fluorescence values of NAO intensity in this period of insult. Data are represented as the mean ± SEM of five independent experiments. Control (452.20 ± 22.28); OGD/R (330.42 ± 23.45); OGD/R + 100 nM raloxifene (404.71 ± 12.34); OGD/R + 10 nM raloxifene (374.64 ± 19.78). Data were examined by analysis of variance, followed by the post hoc Dunnet’s test for between-group comparisons and Tukey’s test for multiple comparisons, * *p* < 0.005, ** *p* < 0.01, **** *p* < 0.0001. Scale bar 50 µm.

## Data Availability

Research data are available and shared in this study.

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
