# Peer review of "Raloxifene Protects Oxygen-Glucose-Deprived Astrocyte Cells Used to Mimic Hypoxic-Ischemic Brain Injury"

_ijms, 2024, doi:10.3390/ijms252212121_

Round 1
Reviewer 1 Report
Comments and Suggestions for Authors
In the work titled “Raloxifene protects oxygen-glucose-deprived astrocyte cells used to mimic hypoxic-ischemic brain injury”, Toro-Urrego and colleagues assessed the protective effect of Raloxifene in an oxygen-glucose deprivation/reoxygenation astrocyte cell model using T98G cell line. The authors showed that Raloxifene is able to improve cell viability, to reduce oxidative stress and to attenuate mitochondrial membrane potential loss and mass reduction. The authors state that these evidence are in line with their previous work where the same human astrocyte cell model was pretreated with Raloxifene and subjected to glucose-deprivation (Raloxifene attenuates oxidative stress and preserves mitochondrial function in astrocytic cells upon glucose deprivation. Vesga-Jiménez et al., 2019).
Overall, the authors presented an interesting follow-up work in this cell model about the neuroprotective role of Raloxifene in an oxygen-glucose deprivation/reoxygenation setup. However, there are some important concerns that need to be assessed before the manuscript acceptance.
Major comments
The authors performed most of their experiments on T98G cells exposed to 6 hours of oxygen and glucose deprivation (OGD), followed by 3 hours of reoxygenation. However, in “Figure 3” and “Figure 5” they assessed different reoxygenation times without providing any explanations about this choice. How do they choose OGD and reoxygenation durations? The data supporting these choices need to be provided as “Supplementary Materials” instead of “data not shown” (as partially indicated in line 205-208). Can the authors provide any data about other reoxygenation timing also for the other experiments, such as cell viability, cell morphology, oxygen superoxide production and mitochondrial membrane potential evaluation?
The authors decided to follow an OGD protocol for their experiments using glucose-free DMEM media. However, the authors applied a “Glucose deprivation” protocol using balanced salt solution free of glucose (BSS0) in their previous work, which provides instrumental data and pivotal working hypothesis for this current work (Raloxifene attenuates oxidative stress and preserves mitochondrial function in astrocytic cells upon glucose deprivation. Vesga-Jiménez et al., 2019). Can the authors explain why they chose to follow this protocol instead of using BSS0 for OGD which would have provided more consistent and robust results?
In line 121, the authors stated that Raloxifene was tested at 3 different concentrations, but data related to 1nm or 10 nm Raloxifene were not always shown. Please provide the missing data (i.e. quantification and microphotographs of viability test, DHE, DCFDA, NAO and TMRM measurements) , at least as Supplementary materials.
In the present manuscript, Raloxifene was added during OGD and reperfusion period. In order to increase the rationale for the use of Raloxifene as treatment in PA, the authors should analyze the effect of the neurosteroid by adding it only in the reperfusion time, mimicking the in vivo situation.
In “Figure 1”, the authors evaluated the morphological changes in T98G cells, providing qualitative evaluations of the difference between cells from “Control”, “OGD/R” and “OGD/R” followed by the respective Raloxifene treatment. As correctly reported by the authors, cells’ viability and morphology are directly related and it is important to assess these aspects, especially in an experimental setup that includes insults in presence or not of a specific treatment. In order to corroborate the effects determined by “OGD/R” and Raloxifene treatment, the authors should provide a quantitative analysis of morphological alterations as described and performed in a previous study (Tibolone protects T98G cells from glucose deprivation. Rodriguez et al., 2014).
The authors stated that DCFDA is oxidized by hydrogen peroxide and other reactive oxygen species (line 243), but in the rest of the manuscript DCFDA-derived fluorescence intensity is considered as an indicator of H2O2 production, and not of ROS production. Similarly, the evaluation of NAO fluorescence is indicated as a marker of mitochondrial mass and of mitochondrial lipid peroxidation. These points need to be better describe and argumented in the text.
Moreover, the description of the results regarding figure 2,3,4,5 appear not clear. In particular, the quantitative vs qualitative data, as well as the origin of the quantification (FACS vs image quantification) are very difficult to distinguish and understand. Please, revise these sections in order to simplify the comprehension of the data.
Minor comments
The abstract should be revised because it should not contain headings, as described in the “Instructions for Authors” guidelines. Moreover, it is difficult to read and follow. Please, revise the abstract following the instructions reported on the website.
The authors did not describe what Raloxifene is in the “Introduction” section and the reasons for its application as a treatment in the context of hypoxic-ischemic brain injury and perinatal asphyxia. Please, add a specific paragraph describing the aforementioned aspects of Raloxifene.
As described in the “Instructions for Authors” guidelines, the “Material and Methods” section must appear after “Results” and “Discussion” sections. Please, revise this part following the instructions reported on the website.
In the “Material and Methods” section, the subsections should follow the same order of the results section (i.e. ROS determination should appear before TMRM methods). Moreover, the FACS measurements of NAO;DCFDA;DHE ; TMRM should be better specify and more detailed.
In the “Material and Methods” section, line 154-157, the description of the CCCP data should be oved in the “Results” section, where these information are missing.
In the “Results” section, the “3.1 Cell viability” and “3.2 Cell morphology” points should be collapsed in a unique paragraph describing the effect of Raloxifene on the viability of T98G cells exposed to OGD/R insults. It is the reviewer’s opinion that this adaptation will help to better understand the results described in Figure 1. Please, see “Melatonin Inhibits Hypoxia-Induced Alzheimer’s Disease Pathogenesis by Regulating the Amyloidogenic Pathway in Human Neuroblastoma Cells” by Singrang and colleagues as an example for describing cell viability data that include cell morphology observations.
All the figure captions should report the statistical tests applied for data analysis, even if this is described in the “Statistical analysis” paragraph of “Material and Methods” section.
The conditions such as “Control”, “OGD”, “OGD/R”, “OGD/R” with the relative Raloxifene treatments, are represented with different colors between figures, making it difficult to understand the graphs. Please, be consistent with the color choice and visual representation.
In the “Figure 1”, the authors should replace the representative microphotographs because it is difficult to appreciate the difference between conditions due to the different background and light exposure, especially in the “Figure 1D”.
In the “Figure 2”, the authors should replace the representative fluorescence micrographs because it is difficult to appreciate any signal of DHE staining. Moreover, they should specify which Raloxifene treatment corresponds to the representative flow cytometry plot reported and they should include the one that is missing, as well as for the fluorescence micrographs. Finally, the authors should provide representative flow cytometry plots of better quality because these are too small and it is difficult to understand the graphs.
In the “Figure 4”, the authors specify which Raloxifene treatment corresponds to the representative flow cytometry plot reported and they should include the one that is missing, as well as for the fluorescence micrographs. Moreover, the authors should provide representative flow cytometry plots of better quality because these are too small and it is difficult to understand the graphs.
Comments on the Quality of English Language
Overall, the quality of the language should be improved, especially for the “Discussion” section. Please, revise the manuscript thoroughly and check for typos, punctuation, abbreviations and spelling errors in order to
Author Response
Note from the Authors. The changes made to the original manuscript appear highlighted in green in the revised version.
Reviewer 1
|
Yes |
Can be improved |
Must be improved |
Not applicable |
|
|
Does the introduction provide sufficient background and include all relevant references? |
( ) |
( ) |
(x) |
( ) |
|
Is the research design appropriate? |
( ) |
( ) |
(x) |
( ) |
|
Are the methods adequately described? |
( ) |
( ) |
(x) |
( ) |
|
Are the results clearly presented? |
( ) |
( ) |
(x) |
( ) |
|
Are the conclusions supported by the results? |
( ) |
( ) |
(x) |
( ) |
Comments and Suggestions for Authors
In the work titled “Raloxifene protects oxygen-glucose-deprived astrocyte cells used to mimic hypoxic-ischemic brain injury”, Toro-Urrego and colleagues assessed the protective effect of Raloxifene in an oxygen-glucose deprivation/reoxygenation astrocyte cell model using T98G cell line. The authors showed that Raloxifene is able to improve cell viability, to reduce oxidative stress and to attenuate mitochondrial membrane potential loss and mass reduction. The authors state that these evidence are in line with their previous work where the same human astrocyte cell model was pretreated with Raloxifene and subjected to glucose-deprivation (Raloxifene attenuates oxidative stress and preserves mitochondrial function in astrocytic cells upon glucose deprivation. Vesga-Jiménez et al., 2019).
Overall, the authors presented an interesting follow-up work in this cell model about the neuroprotective role of Raloxifene in an oxygen-glucose deprivation/reoxygenation setup. However, there are some important concerns that need to be assessed before the manuscript acceptance.
Major comments
- The authors performed most of their experiments on T98G cells exposed to 6 hours of oxygen and glucose deprivation (OGD), followed by 3 hours of reoxygenation. However, in “Figure 3” and “Figure 5” they assessed different reoxygenation times without providing any explanations about this choice. How do they choose OGD and reoxygenation durations? The data supporting these choices need to be provided as “Supplementary Materials” instead of “data not shown” (as partially indicated in line 205-208). Can the authors provide any data about other reoxygenation timing also for the other experiments, such as cell viability, cell morphology, oxygen superoxide production and mitochondrial membrane potential evaluation?
Authors reply. The justification is based on the time interval when reactive species production typically increases, namely after 6 hours of reoxygenation. There are no experiments carried out for other reperfusion times. In similar models, like glucose deprivation, this production has been reported to increase within that time frame in the injury phase (see: https://doi.org/10.3389/fnagi.2016.00152 toro-urrego 21026).
Likewise, the following article is clear with the production of ROS in these time intervals, using markers similar to those analyzed in our research:
Li Y, Wang Y, Yang W, Wu Z, Ma D, Sun J, Tao H, Ye Q, Liu J, Ma Z, Qiu L, Li W, Li L and Hu M (2023) ROS-responsive exogenous functional mitochondria can rescue neural cells post-ischemic stroke. Front. Cell Dev Biol 11:1207748. doi:10.3389/fcell.2023.1207748
In this other article, the authors also report ROS production and mitochondrial membrane potential at different hours, seeing how ROS production and membrane potential are affected after the first hours. This would also justify the different times shown in Figure 5.
Guo, H., Kong, S., Chen, W., Dai, Z., Lin, T., Su, J., … Lai, X. (2014). Apigenin Mediated Protection of OGD-Evoked Neuron-Like Injury in Differentiated PC12 Cells. Neurochemical Research, 39(11), 2197–2210. doi:10.1007/s11064-014-1421-0
- The authors decided to follow an OGD protocol for their experiments using glucose-free DMEM media. However, the authors applied a “Glucose deprivation” protocol using balanced salt solution free of glucose (BSS0) in their previous work, which provides instrumental data and pivotal working hypothesis for this current work (Raloxifene attenuates oxidative stress and preserves mitochondrial function in astrocytic cells upon glucose deprivation. Vesga-Jiménez et al., 2019). Can the authors explain why they chose to follow this protocol instead of using BSS0 for OGD which would have provided more consistent and robust results?
Authors reply. The DMEM model without glucose has been widely used in OGD/R models, supporting a strong basis for its use and implementation. The work by Vesga Jimenez was not ours. Here, a few articles illustrating as said:
Chen J, Sun L, Ding GB, Chen L, Jiang L, Wang J, Wu J. Oxygen-Glucose Deprivation/Reoxygenation Induces Human Brain Microvascular Endothelial Cell Hyperpermeability Via VE-Cadherin Internalization: Roles of RhoA/ROCK2. J Mol Neurosci. 2019 Sep;69(1):49-59. doi: 10.1007/s12031-019-01326-8. Epub 2019 Jun 11. PMID: 31187440.
He, G., Xu, W., Tong, L., Li, S., Su, S., Tan, X., & Li, C. (2016). Gadd45b prevents autophagy and apoptosis against rat cerebral neuron oxygen-glucose deprivation/reperfusion injury. Apoptosis, 21(4), 390–403. doi:10.1007/s10495-016-1213-x
Li, H., Wu, Y., Suo, G., Shen, F., Zhen, Y., Chen, X., & Lv, H. (2018). Profiling neuron-autonomous lncRNA changes upon ischemia/reperfusion injury. Biochemical and Biophysical Research Communications, 495(1), 104–109. doi:10.1016/j.bbrc.2017.10.157 10.1016/j.bbrc.2017.10.157
Xiuwen Kang, Lei Jiang, Xufeng Chen, Xi Wang, Shuangshuang Gu, Jun Wang, Yuanhui Zhu, Xuexue Xie, Hang Xiao, Jinsong Zhang. Exosomes derived from hypoxic bone marrow mesenchymal stem cells rescue OGD-induced injury in neural cells by suppressing NLRP3 inflammasome-mediated pyroptosis,Experimental Cell Research,Volume 405, Issue 1,2021,112635,ISSN 0014-4827, https://doi.org/10.1016/j.yexcr.2021.112635.
Yang Yu, Xiuquan Wu, Jingnan Pu, Peng Luo, Wenke Ma, Jiu Wang, Jialiang Wei, Yuanxin Wang, Zhou Fei. Lycium barbarum polysaccharide protects against oxygen glucose deprivation/reoxygenation-induced apoptosis and autophagic cell death via the PI3K/Akt/mTOR signaling pathway in primary cultured hippocampal neurons https://doi.org/10.1016/j.bbrc.2017.11.165
- In line 121, the authors stated that Raloxifene was tested at 3 different concentrations, but data related to 1nm or 10 nm Raloxifene were not always shown. Please provide the missing data (i.e. quantification and microphotographs of viability test, DHE, DCFDA, NAO and TMRM measurements) , at least as Supplementary materials.
Authors reply. We apologize, there was a mistake in the original manuscript. Only two raloxifene concentrations — 10nM and 100nM — were tested. Figure 3 has been corrected and shows the correct and complete information about the two concentrations.
- In the present manuscript, Raloxifene was added during OGD and reperfusion period. In order to increase the rationale for the use of Raloxifene as treatment in PA, the authors should analyze the effect of the neurosteroid by adding it only in the reperfusion time, mimicking the in vivo situation.
Authors reply. Applied during the reoxygenation period, raloxifene did not reduce OGD-induced cell death. T98G cells were treated with different concentrations of raloxifene during 3 h of reoxygenation, and cell viability was assessed by MTT assay. Bars and error bars represent the mean +/- SEM of 4 independent experiments calculated by 1-way ANOVA tests followed by post hoc multiple comparisons using Tukey’s test. *P < 0.05, Control vs. OGD/R. *P < 0.05 Control vs. OGD/R + 10 nM Ral. *P < 0.05 Control vs. OGD/R + 100 nM Ral. P˃0.05, OGD/R vs. OGD/R + 10 nM Ral. P˃0.05, OGD/R vs. OGD/R + 100 nM Ral. N.S. = No significant differences = P˃0.05.

We believe that viability was not affected post-treatment — shown in the figure — due to the short time between raloxifene administration and measurement. Raloxifene neuroprotection might be exerted at the genomic level, involving its binding to estrogen receptors. Such mechanisms of neuroprotection have been observed beyond 3 hours of treatment. We endorse further experiments to address the participation of estrogen receptors in this model to test this hypothesis.
Veenman, L. Raloxifene as Treatment for Various Types of Brain Injuries and Neurodegenerative Diseases: A Good Start. Int. J. Mol. Sci. 2020, 21, 7586. https://doi.org/10.3390/ijms21207586.
Carroll, J.S. Mechanisms of oestrogen receptor (ER) gene regulation in breast cancer. Eur. J. Endocrinol. 2016, 175, R41–R49
Cyr, M.; Morissette, M.; Landry, M.; Di Paolo, T. Estrogenic activity of tamoxifen and raloxifene on rat brain AMPA receptors. Neuroreport 2001, 12, 535–539.
Yalcin, A.; Kanit, L.; Durmaz, G.; Sargin, S.; Terek, C.H.; Tanyolac, B. Altered level of apurinic/apyrimidinic endonuclease/redox factor-1 (APE/REF-1) mRNA in the hippocampus of ovariectomized rats treated by raloxifene against kainic acid. Clin. Exp. Pharmacol. Physiol. 2005, 32, 611–614.
YazÄŸan, B.; YazÄŸan, Y.; Övey, İ.S.; NazıroÄŸlu, M. Raloxifene and Tamoxifen Reduce PARP Activity, Cytokine and Oxidative Stress Levels in the Brain and Blood of Ovariectomized Rats. J. Mol. Neurosci. 2016, 60, 214–222.
Herceg, Z.; Wang, Z.Q. Functions of poly(ADP-ribose) polymerase (PARP) in DNA repair, genomic integrity and cell death. Mutat. Res. 2001, 477, 97–110.
- In “Figure 1”, the authors evaluated the morphological changes in T98G cells, providing qualitative evaluations of the difference between cells from “Control”, “OGD/R” and “OGD/R” followed by the respective Raloxifene treatment. As correctly reported by the authors, cells’ viability and morphology are directly related and it is important to assess these aspects, especially in an experimental setup that includes insults in presence or not of a specific treatment. In order to corroborate the effects determined by “OGD/R” and Raloxifene treatment, the authors should provide a quantitative analysis of morphological alterations as described and performed in a previous study (Tibolone protects T98G cells from glucose deprivation. Rodriguez et al., 2014).
Authors reply. Quantification was not done previously, as we considered the differences were visible in the images. The revised manuscript includes quantification following your suggestion (Rodriguez et al., 2014). Additional information is provided under materials and methods, and results.:
Materials and Methods
The mean cell area was assessed by analyzing black-and-white phase contrast microphotographs using ImageJ software, version 1.54j. The area of each randomly selected cell was determined, and the software was calibrated by measuring a known distance. The average cell area was determined for every experimental group, each measured in triplicate, with at least 25 cells analyzed per condition.
Results
Our subsequent objective was to analyze the average cell area of randomly chosen cells within our experimental settings. We qualitatively noted a change in cell morphology during glucose deprivation (OGD/R) stress (Fig. 1C). Post-Raloxifene co-treatment, the disruptions in morphology were reduced, returning to their control levels. To quantify morphological changes, we assessed the mean cell area across all experimental conditions. Raloxifene-treated cells (cell area 450 ± 35 µm) were comparable to control cell areas (cell area 460 ± 22 µm) (P < 0.0019) while a decrease was observed in the OGD/R group compared with control cells (cell area 311 ± 37 µm) (P < 0.5654).
- The authors stated that DCFDA is oxidized by hydrogen peroxide and other reactive oxygen species (line 243), but in the rest of the manuscript DCFDA-derived fluorescence intensity is considered as an indicator of H2O2 production, and not of ROS production. Similarly, the evaluation of NAO fluorescence is indicated as a marker of mitochondrial mass and of mitochondrial lipid peroxidation. These points need to be better describe and argumented in the text.
Authors reply. The DCFDA marker is oxidized by peroxide as well as other oxidizing agents present in the cell. However, it is not wrong to speak of a general state of oxidative stress at first glance. Articles can be cited that support this being an indicator of this type:
“Intracellular esterases cleave DCFH-DA at the two ester bonds, producing a relatively polar and cell membrane-impermeable product, H2 DCF. This non-fluorescent molecule accumulates intracellularly and subsequent oxidation yields the highly fluorescent product DCF. The redox state of the sample can be monitored by detecting the increase in fluorescence. Accumulation of DCF in cells may be measured by an increase in fluorescence at 530 nm when the sample is excited at 485 nm. Fluorescence at 530 nm can be measured using a flow cytometer and is assumed to be proportional to the concentration of hydrogen peroxide in the cells” Eruslanov, E., & Kusmartsev, S. (2009). Identification of ROS Using Oxidized DCFDA and Flow-Cytometry. Advanced Protocols in Oxidative Stress II, 57–72. doi:10.1007/978-1-60761-411-1_4
In the following article, they use this probe specifically for measuring H2O2:
“Intracellular ROS measurement
The intracellular H2O2 content of oocytes and putative embryos was measured before and after incubation with sperm in an IVF medium. Oocytes and putative embryos were washed 3 times in phosphate-buffered saline (PBS) and then incubated in PBS supplemented with 10 μm DCFDA (10 ml, Sigma, Life Technologies C6827) for 15 min at 37 °C and subsequently slowly washed with PBS. The stained oocytes or putative embryos were transferred into a droplet on a glass slide and observed using an inverted fluorescence microscope at 488 nm excitation wavelength. Finally, the pixel intensity within recorded fluorescent images was analyzed using ImageJ software”
Navid S, Saadatian Z, Talebi A. Assessment of developmental rate of mouse embryos yielded from in vitro fertilization of the oocyte with treatment of melatonin and vitamin C simultaneously. BMC Womens Health. 2023 Oct 4;23(1):525. doi:10.1186/s12905-023-02673-w. PMID: 37794412; PMCID: PMC10552323.
Another article also talks about this:
Rhee SG, Chang TS, Jeong W, Kang D. Methods for detection and measurement of hydrogen peroxide inside and outside of cells. Mol Cells. 2010 Jun;29(6):539-49. doi:10.1007/s10059-010-0082-3. Epub 2010 Jun 4. PMID: 20526816.
Regarding NAO, it is an indicator of lipid peroxidation, especially of cardiolipin, an internal mitochondrial membrane constituent lipid. It has been widely used to detect lipid peroxidation and measure mitochondrial mass as a marker:
“The high-affinity binding of NAO to CL has been used to determine a number of properties of CL; e.g., to image CL in cells by confocal microscopy (Jacobson et al., 2002), to measure mitochondrial mass per cell (Guidot, 1998), and to quantify the level of CL in the inner and outer leaflets of the mitochondrial inner membrane (Garcia Fernandez et al., 2002) article: Rodriguez ME, Azizuddin K, Zhang P, Chiu SM, Lam M, Kenney ME, Burda C, Oleinick NL. Targeting of mitochondria by 10-N-alkyl acridine orange analogues: role of alkyl chain length in determining cellular uptake and localization. Mitochondrion. 2008 Jun;8(3):237-46. doi: 10.1016/j.mito.2008.04.003. Epub 2008 Apr 25. PMID: 18514589; PMCID: PMC2585370.
- Mitochondrial mass: Guidot DM. Endotoxin pretreatment in vivo increases the mitochondrial respiratory capacity in rat hepatocytes. Arch. Biochem. Biophys. 1998;354:9–17. [PubMed] [Google Scholar]
- Jacobson J, Duchen MR, Heales SJ. Intracellular distribution of the fluorescent dye nonyl acridine orange responds to the mitochondrial membrane potential: implications for assays of cardiolipin and mitochondrial mass. J. Neurochem. 2002;82:224–233.
- Garcia Fernandez M, Troiano L, Moretti L, Nasi M, Pinti M, Salvioli S, Dobrucki J, Cossarizza A. Early changes in intramitochondrial cardiolipin distribution during apoptosis. Cell Growth Differ. 2002;13:449–455.
- Jacobson J, Duchen MR, Heales SJ. Intracellular distribution of the fluorescent dye nonyl acridine orange responds to the mitochondrial membrane potential: implications for assays of cardiolipin and mitochondrial mass. J Neurochem. 2002 Jul;82(2):224-33. doi: 10.1046/j.1471-4159.2002.00945.x. PMID: 12124423.
- Maftah A, Petit JM, Ratinaud MH, Julien R. 10-N nonyl-acridine orange: a fluorescent probe which stains mitochondria independently of their energetic state. Biochem Biophys Res Commun. 1989 Oct 16;164(1):185-90. doi: 10.1016/0006-291x(89)91700-2. PMID: 2478126.
While these terms seem appropriate to talk about mitochondrial mass or lipid peroxidation, we have used lipid peroxidation, to unify the language used in the text.
- Moreover, the description of the results regarding figure 2,3,4,5 appear not clear. In particular, the quantitative vs qualitative data, as well as the origin of the quantification (FACS vs image quantification) are very difficult to distinguish and understand. Please, revise these sections in order to simplify the comprehension of the data.
Authors reply. This point has been clarified in Materials and Methods
Minor comments
- The abstract should be revised because it should not contain headings, as described in the “Instructions for Authors” guidelines. Moreover, it is difficult to read and follow. Please, revise the abstract following the instructions reported on the website.
Authors reply. Thank you for your observation. The abstract has been corrected and revised, following your instructions and those reported on the website.
- The authors did not describe what Raloxifene is in the “Introduction” section and the reasons for its application as a treatment in the context of hypoxic-ischemic brain injury and perinatal asphyxia. Please, add a specific paragraph describing the aforementioned aspects of Raloxifene.
Authors reply. The information requested has been added in the introduction. We appreciate your suggestion.
- As described in the “Instructions for Authors” guidelines, the “Material and Methods” section must appear after “Results” and “Discussion” sections. Please, revise this part following the instructions reported on the website.
Authors reply. Sections have been reordered, following the journal’s instructions.
- In the “Material and Methods” section, the subsections should follow the same order of the results section (i.e. ROS determination should appear before TMRM methods). Moreover, the FACS measurements of NAO; DCFDA; DHE; TMRM should be better specify and more detailed.
Authors reply. Subsections have been reordered. The Material and methods section has been improved with more specificity and details.
- In the “Material and Methods” section, line 154-157, the description of the CCCP data should be oved in the “Results” section, where these information are missing.
Authors reply. This information has been added in the Results section.
- In the “Results” section, the “3.1 Cell viability” and “3.2 Cell morphology” points should be collapsed in a unique paragraph describing the effect of Raloxifene on the viability of T98G cells exposed to OGD/R insults. It is the reviewer’s opinion that this adaptation will help to better understand the results described in Figure 1. Please, see “Melatonin Inhibits Hypoxia-Induced Alzheimer’s Disease Pathogenesis by Regulating the Amyloidogenic Pathway in Human Neuroblastoma Cells” by Singrang and colleagues as an example for describing cell viability data that include cell morphology observations.
Authors reply. The information has been unified and clarified in the Results section following your suggestion.
- All the figure captions should report the statistical tests applied for data analysis, even if this is described in the “Statistical analysis” paragraph of “Material and Methods” section.
Authors reply. The statistical tests used to analyze data have been added to the figure caption. All the bar graphs have been colored consistently, and Figure 3 shows the missing group along with its statistical description.
- The conditions such as “Control”, “OGD”, “OGD/R”, “OGD/R” with the relative Raloxifene treatments, are represented with different colors between figures, making it difficult to understand the graphs. Please, be consistent with the color choice and visual representation.
Authors reply. All the bar graphs have been colored consistently.
- In the “Figure 1”, the authors should replace the representative microphotographs because it is difficult to appreciate the difference between conditions due to the different background and light exposure, especially in the “Figure 1D”.
Authors reply. The original figure has been replaced with a better-quality one.
- In the “Figure 2”, the authors should replace the representative fluorescence micrographs because it is difficult to appreciate any signal of DHE staining. Moreover, they should specify which Raloxifene treatment corresponds to the representative flow cytometry plot reported and they should include the one that is missing, as well as for the fluorescence micrographs. Finally, the authors should provide representative flow cytometry plots of better quality because these are too small and it is difficult to understand the graphs.
Authors reply. Figure 2 has been replaced with an improved one, and the missing photograph has been added. Also, better flow cytometry plots have been built.
- In the “Figure 4”, the authors specify which Raloxifene treatment corresponds to the representative flow cytometry plot reported and they should include the one that is missing, as well as for the fluorescence micrographs. Moreover, the authors should provide representative flow cytometry plots of better quality because these are too small and it is difficult to understand the graphs.
Authors reply. Figure 4 has been replaced with an improved one, and the missing photograph has been added. Also, better flow cytometry plots have been built.
- Comments on the Quality of English Language. Overall, the quality of the language should be improved, especially for the “Discussion” section. Please, revise the manuscript thoroughly and check for typos, punctuation, abbreviations and spelling errors in order to.
Authors reply. The manuscript underwent a thorough language revision by a native speaker to improve the overall quality, especially in the “Discussion” section. Typos, punctuation, abbreviations, and spelling errors were checked and corrected.

Reviewer 2 Report
Comments and Suggestions for Authors
Dear authors, thank you very much for this interesting report, which certainly shows a new and promising results of Raloxifene on hypoxic-ishemic brain injury. The aim of this preclinical study was to investigate whether raloxifene showed neuroprotection in an oxygen-glucose deprivation/reoxygenation astrocyte cell model.
To improve your manuscript, I suggest the following changes that should be made by the authors:
- - The introductory section provided a good insight into the known facts describing the ethiopathogenesis of perinatal asphyxia (PA), as well the importance of the astrocytes for brain glucose metabolism. It would be interesting to briefly include basic information on the use, content and mechanism of action of raloxifene in the introduction.
- The M&M section is very well described, but I would like to know if the qualitative assessment is done by one or more examiners?
I look forward to your reply. Kind regards!
Comments on the Quality of English LanguageNo comments.
Author Response
Note from the Authors. The changes made to the original manuscript appear highlighted in green in the revised version.
Reviewer 2
|
|
||||||
|
Yes |
Can be improved |
Must be improved |
Not applicable |
|
||
|
Does the introduction provide sufficient background and include all relevant references? |
(x) |
( ) |
( ) |
( ) |
|
|
|
Is the research design appropriate? |
( ) |
(x) |
( ) |
( ) |
|
|
|
Are the methods adequately described? |
( ) |
(x) |
( ) |
( ) |
|
|
|
Are the results clearly presented? |
( ) |
(x) |
( ) |
( ) |
|
|
|
Are the conclusions supported by the results? |
(x) |
( ) |
( ) |
( ) |
|
|
Comments and Suggestions for Authors
Dear authors, thank you very much for this interesting report, which certainly shows a new and promising results of Raloxifene on hypoxic-ischemic brain injury. The aim of this preclinical study was to investigate whether raloxifene showed neuroprotection in an oxygen-glucose deprivation/reoxygenation astrocyte cell model.
To improve your manuscript, I suggest the following changes that should be made by the authors:
- The introductory section provided a good insight into the known facts describing the ethiopathogenesis of perinatal asphyxia (PA), as well the importance of the astrocytes for brain glucose metabolism. It would be interesting to briefly include basic information on the use, content and mechanism of action of raloxifene in the introduction.
Authors reply. The information requested has been added in the introduction. We appreciate your suggestion.
- The M&M section is very well described, but I would like to know if the qualitative assessment is done by one or more examiners?
Authors reply. Two examiners performed the qualitative assessment in a double-blind fashion.
Comments on the Quality of English Language: No issues detected.
__________________________
Round 2
Reviewer 1 Report
Comments and Suggestions for Authors
I acknowledge and welcome the efforts from Nicolás Toro-Urrego and colleagues to address the referees’ comments about their work titled “Raloxifene protects oxygen-glucose-deprived astrocyte cells used to mimic hypoxic-ischemic brain injury”. It is the referee’s opinion that the authors made the manuscript gain scientific value and impact by addressing the referees’ concerns. However, there are still some important points to address prior to publication.
Most of the figures presented in the manuscript are still difficult to read and interpret correctly. The main flaw is the order in which the representative microphotographs appear relative to the data presented in the graph. For example, in Figure 1A the order of experimental groups is: CTRL, OGD/R, OGD/R + Ral 100 nM and OGD/R + Ral 10 nM. However, representative microphotographs are misplaced and they do not follow the same left-to-right orientation. This aspect leads to a difficult reading and interpretation of the graph. Moreover, Figure 1 microphotographs should be named from 1B to 1E, not 1F. Similarly, Figure 5 labeling is skipping letter M and N. Furthermore, all the microphotographs should have the experimental group label. For example, Figures 3A and 3B do not have their label. Please, revise all the figures within the manuscript in order to improve their understanding.
The authors successfully demonstrated that OGD/R followed by Raloxifene treatment did not rescue T98G viability, as reported in the cover letter. This evidence lays the foundation for exploring the molecular mechanisms linked to its neuroprotective effect, as well as providing important insights related to pharmacological intervention in the context of perinatal asphyxia. Based on the importance of this observation, it is the referee’s opinion that this data should be included and properly discussed in the manuscript, together with the quantitative analysis of morphological alterations of T98G cells treated with Raloxifene after OGD/R.
Since the authors assessed morphological alterations by quantitative analysis, it is the referee’s opinion that this data should be displayed in the manuscript because it further proves that Raloxifene treatment is able to improve cell vitality during OGD/R (for example, this graph should be placed in Figure 1).
It’s still wrongly reported that T98G cells were treated with three different concentrations of Raloxifene (for example, line 26 and line 427). Please, check the manuscript in order to correct this aspect since just two concentrations were tested, as answered by the authors in the cover letter.
Comments on the Quality of English Languagethe quality of english language is sufficient.
Author Response
Reviewer comment
Most of the figures presented in the manuscript are still difficult to read and interpret correctly. The main flaw is the order in which the representative microphotographs appear relative to the data presented in the graph. For example, in Figure 1A the order of experimental groups is: CTRL, OGD/R, OGD/R + Ral 100 nM and OGD/R + Ral 10 nM. However, representative microphotographs are misplaced and they do not follow the same left-to-right orientation. This aspect leads to a difficult reading and interpretation of the graph. Moreover, Figure 1 microphotographs should be named from 1B to 1E, not 1F. Similarly, Figure 5 labeling is skipping letter M and N. Furthermore, all the microphotographs should have the experimental group label. For example, Figures 3A and 3B do not have their label. Please, revise all the figures within the manuscript in order to improve their understanding.
Authors reply. Thank you for your comments and suggestions. We modified the figures and their respective legends accordingly.
Reviewer comment
The authors successfully demonstrated that OGD/R followed by Raloxifene treatment did not rescue T98G viability, as reported in the cover letter. This evidence lays the foundation for exploring the molecular mechanisms linked to its neuroprotective effect, as well as providing important insights related to pharmacological intervention in the context of perinatal asphyxia. Based on the importance of this observation, it is the referee’s opinion that this data should be included and properly discussed in the manuscript, together with the quantitative analysis of morphological alterations of T98G cells treated with Raloxifene after OGD/R.
Authors reply. Raloxifene did not change cell viability in this model when administered solely during reoxygenation, as addressed in previous revisions. In this study, raloxifene successfully reduced cell death in our astrocytic OGD model (Figure 1). Raloxifene treatment protected T98G cells from viability loss and improved several mitochondrial parameters, posing raloxifene as an effective treatment for OGD/R injury in these conditions. Our results oppose the reviewer’s comment that raloxifene did not rescue T98G viability.
Regarding the implications of raloxifene on morphological changes and its relationship with cell viability (Figure 1), cells treated with raloxifene were not different from those in the OGD group. Future experiments are recommended to elucidate raloxifene effects on cell morphology in this model. Raloxifene’s effects on cell morphology are well-established in different cell types, involving complex molecular mechanisms.
Observed Morphological Changes:
Raloxifene induces a series of morphological changes in neuronal and glial cells, including alterations in cell size and shape, as well as changes in cytoskeletal structure. These changes translate into improved cell survival and synaptic plasticity, which are crucial factors for the treatment of neurodegenerative diseases.
Metabolic Pathways and Involved Proteins:
Estrogen Receptor (ER) Pathway:
ERα and ERβ Receptors: Raloxifene binds to these receptors, activating the transcription of genes involved in cell proliferation and differentiation. This binding also modulates apoptosis, contributing to neuroprotection.
Genomic and Non-Genomic Pathways: The activation of ERs by raloxifene triggers both genomic and non-genomic effects, influencing gene expression and intracellular signaling cascades.
PI3K/AKT Pathway:
Raloxifene activates AKT phosphorylation, promoting cell survival and growth. This pathway is involved in regulating cell morphology through modulation of the cytoskeleton and cell adhesion.
MAPK/ERK Pathway:
Activation of the MAPK/ERK cascade by raloxifene results in the phosphorylation of key proteins that regulate the cell cycle and morphology, including transcription factors that promote cell differentiation and proliferation.
JAK/STAT Pathway:
Raloxifene may influence the JAK/STAT pathway, modulating the expression of genes involved in inflammatory responses and cell growth. This pathway is essential for mediating the anti-inflammatory and neuroprotective effects of raloxifene.
Involved Proteins and Receptors:
Estrogen Receptors (ERα and ERβ): Principal mediators of raloxifene's effects.
AKT (Protein Kinase B): Involved in cell survival and growth.
ERK (Extracellular Signal-Regulated Kinase): Regulates cell differentiation and proliferation.
STAT (Signal Transducer and Activator of Transcription): Participates in signal transduction for gene expression regulation.
Detailed Molecular Mechanisms:
Modulation of Genomic Transcription:
Raloxifene binds to estrogen receptors, affecting the transcription of genes such as Bcl-2, Bax, and c-Myc, which are crucial for regulating apoptosis and cell proliferation.
For a more in-depth analysis, refer to Veenman, Leo. 2020. "Raloxifene as Treatment for Various Types of Brain Injuries and Neurodegenerative Diseases: A Good Start" International Journal of Molecular Sciences 21, no. 20: 7586. https://doi.org/10.3390/ijms21207586
Reviewer comment
Since the authors assessed morphological alterations by quantitative analysis, it is the referee’s opinion that this data should be displayed in the manuscript because it further proves that Raloxifene treatment is able to improve cell vitality during OGD/R (for example, this graph should be placed in Figure 1).
Authors reply We have added the quantification graph to figure 1.
Reviewer comment
It’s still wrongly reported that T98G cells were treated with three different concentrations of Raloxifene (for example, line 26 and line 427). Please, check the manuscript in order to correct this aspect since just two concentrations were tested, as answered by the authors in the cover letter
Authors reply Thank you very much for your careful observation. We corrected these errors.
